# Development of Integrated Flooding Early Warning and Rainfall Runoff Management Platform for Downtown Area of Shanghai

**Zhenbao Shi [1,2], Qingran Shen [2], Qiong Tan [3] and Tian Li [1,*]**

[1]   State Key Laboratory of Pollution Control and Resource Reuse, Tongji University, Shanghai 200092, China; shizhenbao2@163.com
[2]   Shanghai Bibo Water Design and Research Center, Shanghai 200233, China; ndiwill@163.com
[3]   Shanghai Water Planning and Design Research Institute, Shanghai 200233, China; tanq20@163.com
[*]   Correspondence: tianli@tongji.edu.cn

**Abstract:** To enhance the capacity of Shanghai's drainage network to guard against flooding, this study used data obtained from an urban drainage network and spatial geological information to conduct precise analysis on an area of approximately 31 km$^2$ with various land uses in downtown Shanghai and to establish a two-dimensional model. Based on the two-dimensional model, an integrated urban flooding early warning and rainfall runoff management platform was developed through combining meteorological data and real-time remote sensing data of the drainage network operation. Through precise simulation of the rainstorm runoff process, projection of the scope and magnitude of urban surface runoff hazard impact, issuance of flooding forecasts, and provision of hazard early warning and decision-making support, the developed platform is capable of providing risk assessment of the drainage system and early warning of flooding risk.

**Keywords:** drainage network; real-time model; flooding early warning; Smart Water; risk assessment



## 1. Introduction

In recent years, flooding caused by extreme rainstorm events has been of worldwide concern [1,2]. It has been recognized that change is required in relation to the prevention and mitigation of the urban flooding hazard and that there is an increasing need for greater public safety. Consequently, many countries have changed their approach from the concept of "flood control" to "flooding management", and they have adopted "flooding risk management" models to minimize the flood hazard impact by leveraging both engineering and non-engineering practices for flood hazard mitigation. For example, the UK has implemented an integrated scheduling system that encompasses risk assessment, flood hazard early warning, emergency planning, land use planning, and flood hazard prevention and control. After the occurrence of severe flooding in 2007, the UK conducted a post event evaluation [3] and the Flood and Water Management Act was issued in 2010 [as of 2010, the Legislation.Gov.UK listed on its website http://www.legislation.gov.uk/ukpga/2010/29/contents (accessed on 30 September 2021).]. This Act identifies the key role of early warning modeling technology in relation both to flooding management and to continuous improvement of the quality of flood hazard forecasting. The USA focuses its efforts on flood hazard forecasting and early warning through the issuance of meteorological, hydrological, and flooding information to the public. Various river flooding forecast centers have been established with forecast lead times of up to hours for central rivers and weeks for major river bodies. The hydrological and meteorological information issued to the public via news media, such as the Internet and television, is used for flood prevention and mitigation as well as economic construction [4]. In Japan, integrated flood control approaches adopted in urban areas include bank regulation, catchment flooding, and flood

hazard mitigation countermeasures. Such mitigation countermeasures mainly comprise warning and evacuation systems, enhanced hazard mitigation practices, publication of flood records and details of areas at risk of flooding, and flood control facility support. In 1994, Japan developed flood maps that allow for the identification of potential flooding areas, evacuation locations, and emergency escape routes [5]. The government sector works with the national meteorological agency to forecast water levels and flows, issue flood warnings, and disseminate such information to the public via the news media. With continuous advancement of the urbanization process and the rapid development of the urban and rural economies in China, flooding events resulting from rainstorms and the scale of related losses have both increased. Since the 1980s, research on flooding in China has considered early warning, forecasting, management, and control with the objective of addressing the flooding hazard from a micro perspective based on hydrological and hydraulic engineering science. It has focused primarily on the following two aspects: (i) monitoring, evaluation, and early warning of the flooding hazard using state-of-the-art information technology, e.g., remote sensing (RS) techniques, geographic information system (GIS) data, and global positioning system (GPS) data [6,7]; (ii) study on flood hazard emergency scheduling based on numerical technological and hydrological models [8].

In China, due to rapid urban development and climate change, extreme meteorological events are occurring more frequently. For example, on July 20th of 2021, more than 200 mm of rain fell on the city of Zhengzhou in a single hour, in China's Henan province. The precipitation accumulated up to 552 mm in 24 h (breaking the historical extreme records), and caused huge losses of personnel and property. However, the current measures to deal with flood disasters are relatively single, and there is a lack of technical means to forecast and warn the areas where serious flooding disasters occur. There is a large gap between the accuracy of disaster early warning and the requirements of safety assurance. It is necessary to improve the ability of flood prevention and disaster mitigation early warning and forecasting and the ability to deal with major flood disasters. In particular, it is necessary to develop a multi-disciplinary, cross-domain, and multi-department integrated management and control platform.

Therefore, the development of a system for urban drainage network management based on real-time monitoring and on an early warning and forecasting model is the main trend in current research for urban flooding prevention and management [9,10]. The implementation of this project fills up the gaps in the application of this field in Chinese cities and towns.

This platform will not only improve the capacity of urban drainage networks to guard against flooding, but also provide a support for drainage network scheduling, flooding early warning, and flood hazard mitigation. On the one hand, the platform can enhance features that include flooding early warning, forecasting, and risk assessment based on numerical models, providing technical support for the safety of drainage and flooding in the study area. On the other hand, the platform has formed a set of key technologies for the construction of a real-time warning model for drainage systems, and provided a blueprint for the construction of similar flood control platforms in other medium-sized and above cities and towns.

## 2. Materials and Methods

### 2.1. Scope of Study

Nine drainage systems in the Xujiahui District of downtown Shanghai were selected for the pilot study (Figure 1, the area is located at 121.45 north latitude and 31.2 east longitude). Covering a serviced area of 31 km$^2$, the pilot study considered 8 combined systems and 1 separate collection system that included 10 flood control drainage pump stations. The area is located in the CBD business district, where the government office and foreign consulate are located. It has a high construction density, a large drainage system, a low drainage standard, and a high risk of water accumulation.

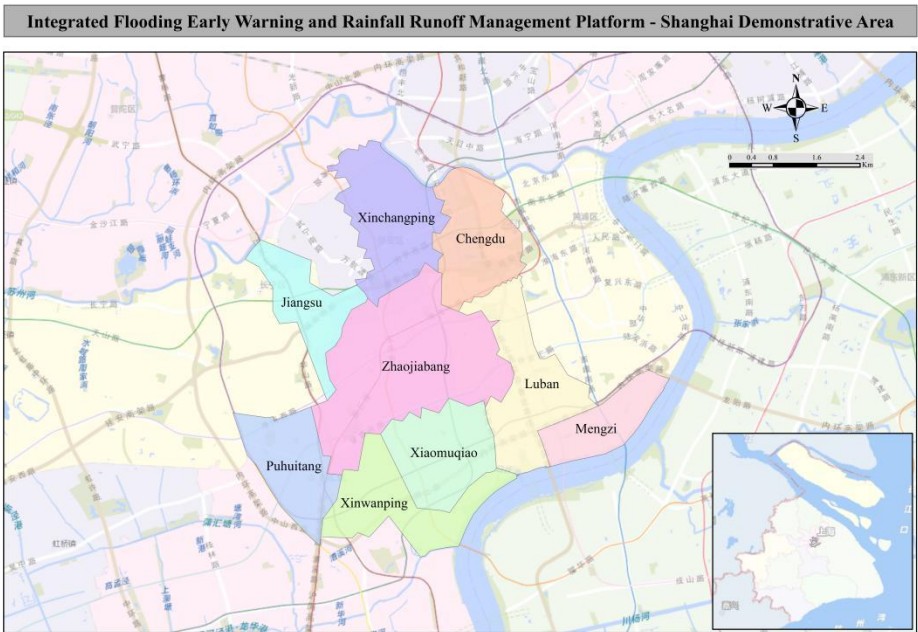

**Figure 1.** Drainage systems in the study area.

### 2.2. Data Collection

Through cooperation with various administrative sectors of Shanghai, a range of data pertaining to the research area were obtained for the development of a sophisticated drainage network model. These data included spatial geological data, rainfall measurements, current drainage facility data, drainage system construction data, drainage facility operation, monitoring, and maintenance data, as well as drainage facility real-time operating data. These real-time data and basic data provided the foundation for the construction of a real-time early warning and forecasting model.

(1) Drainage facility GIS

A drainage facility GIS database built for Shanghai allows real-time acquisition, storage, and access of various data that include drainage pumping station operating data, rainfall data, and water level data at flood monitoring points. The real-time data acquisition interval is 1 min. In addition, the drainage facility GIS includes integrated historical monitoring data for browsing, inquiry, and download.

(2) Spatial geological data

Various layers of information were extracted from the basic topographical map (scale: 1:500) of the research area, e.g., administrative divisions, roads, water systems, buildings, vegetation, and elevation. Where relevant, the data adopt the Shanghai Urban Construction Coordinate System with elevations referred to the Shanghai Wusong datum elevation (Sheshan).

The research area is characterized by terrain that is relatively high in the south and low in the north; however, most of the land surface has an elevation of 3–5 m (based on the Wusong datum elevation). The Zhaojiabang drainage system, located in the center of the research area, is characterized by reasonably flat terrain (elevation: 3–5 m) with a minor gradient of decrease toward the north. Rainwater within the research area is lifted by pumping stations into the watercourse and discharged into the Huangpu River.

Based on up-to-date satellite imagery, the land use of the study area was divided into four categories: roads, roof, greenbelt, and other, which served as the basis for the sophisticated drainage network model simulations.

(3) Rainfall measurements

There are nine rain gauge stations within the research area, i.e., one in each of the drainage systems. The tipping bucket rain gauges automatically measure rainfall at an interval of 1–5 min. Rainfall data were collected for rainstorm events since 2010 that led

to major flooding in Shanghai (Table 1). Cross-correction and compilation of rainfall data from adjacent rain gauge stations were conducted to ensure data consistency, integrity, and accuracy.

**Table 1.** Details of major rainfall events in Shanghai since 2010.

| No. | Date | Accumulated Rainfall (mm) | Maximum Hourly Precipitation (mm/h) |
|---|---|---|---|
| 1 | 3 July 2010 | 78.7–115.3 | 28.1–40.6 |
| 2 | 17 August 2010 | 9.5–65.5 | 9.1–55.2 |
| 3 | 25 August 2010 | 12.7–91.5 | 5.6–43.9 |
| 4 | 1 September 2010 | 58.4–148.6 | 22.4–56.2 |
| 5 | 13 September 2013 | 26.9–115.6 | 21.6–88 |
| 6 | 7–9 October 2013 (Typhoon Fitow) | 145.7–224.1 | 14.6–24 |
| 7 | 17 June 2015 | 131.6–231.0 | 14.4–50.1 |

(4) Current drainage facility data

Data relating to the drainage facilities of the nine drainage systems within the research area were collected to provide basic data for the development of the model (Table 2) and pumping station on/off water level in the research area are shown in Figure 2.

**Table 2.** Data relating to the drainage facilities within the study research area.

| No. | Item | Format | Major Content |
|---|---|---|---|
| 1 | Planning layout, design drawing, as-built drawing | CAD file, images | Construction data of drainage system and supporting facilities |
| 2 | Inspection shaft | Shape file | Location, elevation |
| 3 | Pipeline | Shape file | Location, pipe diameter, elevation |
| 4 | Supporting facilities including pump station, etc. | Shape file, images | Location, pump capacity, number of pumps, head of delivery chart |
| 5 | Pump station scheduling rules | Documents | Water pump on/off water level |

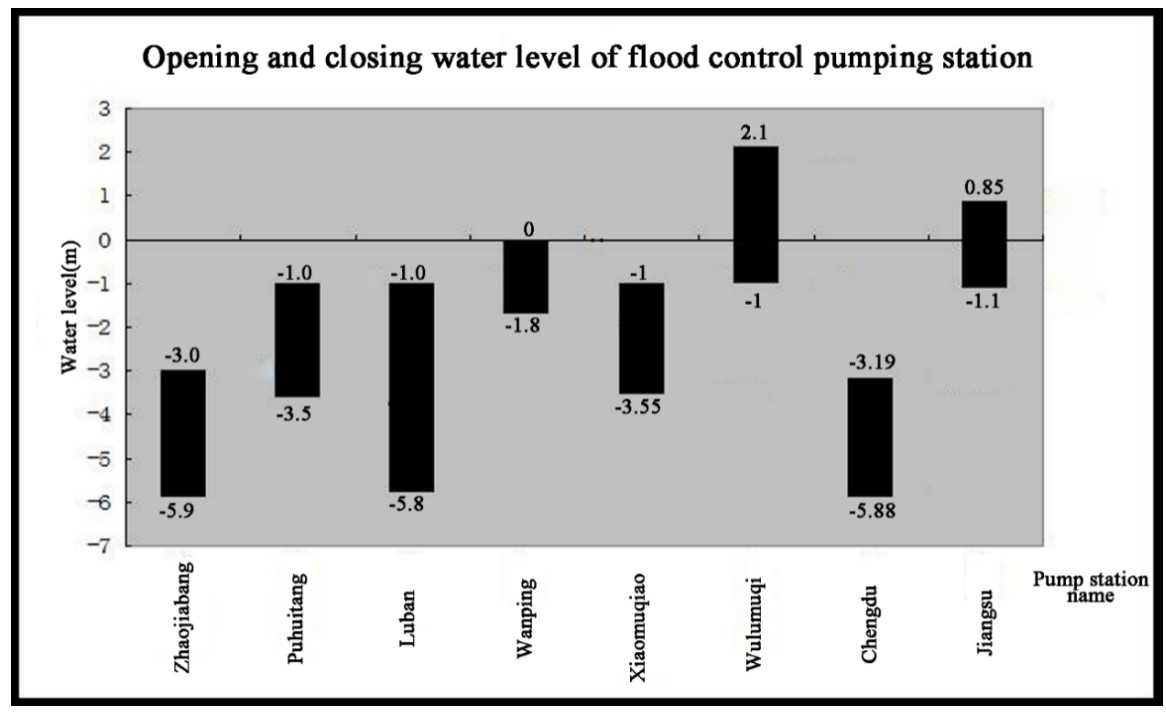

**Figure 2.** Pumping station on/off water level in the research area.

(5) Construction data of drainage system

Construction data of the nine drainage systems within the research area were collected, including the system name, drainage system, service area, integrated runoff coefficient, drainage standard, pump station flow, and receiving river channel name, as shown in Table 3.

**Table 3.** Drainage system construction data.

| No. | Item | System Category | Service Area (km²) | Comprehensive Design Runoff Coefficiency | Current Discharge Standard | Current Pump Station Name | Current Flow m³/s | Planning Flow m³/s | Interception Capacity m³/s | Receiving Water Body |
|---|---|---|---|---|---|---|---|---|---|---|
| 1 | Zhaojiabang | Combined system | 7.38 | 0.67 | 1 | Zhaojiabang | 29.43 | 34.2 | 4.8 | Huangpu River |
| 2 | Puhuitang | Combined system | 2.6 | 0.7 | 1 | Puhuitang | 16.98 | 16.98 | 1.52 | Puhuitang |
| 3 | Luban | Combined system | 3.6 | 0.8 | 1 | Luban | 25.6 | 31 | 5.41 | Huangpu River |
| 4 | Xiaomuqiao | Combined system | 2.93 | 0.6 | 1 | Xiaomuqiao | 18 | 18 | 1.92 | Huangpu River |
| 5 | Xinwanping | Combined system | 3.13 | 0.6 | 1 | Xinwanping | 9.2 | 17.2 | 1.11 | Longhua habour |
| 6 | Jiangsu | Combined system | 2.18 | 0.69 | 1 | Jiangsu | 12.9 | 12.9 | 4.8 | Suzhou River |
| 7 | Chengdu | Combined system | 3.03 | 0.71 | 1 | Chengdu | 22.495 | 25.8 | 3.3 | Suzhou River |
| 8 | Xinchangping | Combined system | 3.66 | 0.7 | 1 | Xinchangping | 19.98 | 22 | 3.01 | Suzhou River |
| 9 | Shibomengzi | Seprate system | 1.88 | 0.6 | 5 | Mengzi | 16.8 | 16.8 | 0.36 | Huangpu River |

(6) Drainage facility operation monitoring and maintenance data

The SCADA (Supervisory Control And Data Acquisition) logs of the drainage pumping stations, flood monitoring points, and CCTV detection data of the drainage pipelines were collected, as shown in Table 4.

**Table 4.** Operation monitoring and maintenance data of the drainage systems within the study area.

| No. | Item | Format | Major Content |
|---|---|---|---|
| 1 | SCADA log of drainage pump station | Database, CSV, txt files, etc. based on interval of 1–5 min | Field rainfall measurements, intake structure water level, pump on/off status of the 9 drainage pump stations within the research area |
| 2 | Flooding monitoring point SCADA log | Database, CSV, txt files, etc. based on interval of 1–5 min | Field water depth measurements of 3–5 flooding monitoring points within the research area |
| 3 | Drainage pipeline CCTV detection report | Txt files | Functionality and structure defective results of drainage pipelines detected by CCTV |

*2.3. Management Platform Architecture*

The integrated urban flooding early warning and rainfall runoff management platform consists of four components: a drainage facility GIS, off-line drainage network model, on-line real-time early warning and forecasting system, and web publishing system for real-time presentation of simulation results.

*2.4. Selection of Modeling Software Platform*

The off-line hydrodynamic modeling software platforms used most often in previous studies include SWMM, InfoWorks CS, InfoWorks ICM, and DHI urban, and the most commonly used on-line real-time early warning modeling software platforms include FloodWorks and MIKE Flood [11]. SWMM cannot simulate two-dimensional flooding. DHI series software is more inclined to watershed simulation. This project needs to carry out two-dimensional real-time simulation of flooding at the level of urban drainage system, therefore the InfoWorks series software was selected as the modeling software platform.

(1) Selection of off-line hydrodynamic modeling software platform

By taking into account the features of existing platforms, and software portability and flexibility, the InfoWorks CS software platform, developed by Wallingford Software (UK), was selected to establish the 1- and 2-dimensional hydrodynamic coupling models in this

study. The InfoWorks CS software has also often been used in drainage network modeling applications in Shanghai.

(2) Selection of on-line real-time early warning modeling software platform

In this study, the FloodWorks software was selected as the on-line real-time early warning modeling software platform. Developed based on sophisticated hydrological and hydraulic models, with features that include real-time analysis and model reports as well as decision-making support and control, FloodWorks offers universal on-line decision-making support for on-line scheduling of urban drainage networks and real-time flood forecasting. Integrating various data acquisition modules (e.g., SCADA) and the InfoWorks simulation engine, FloodWorks features client/server-based architecture and a multiuser and multimodel system structure. The core of the forecasting module is based on the River and Flood Forecasting System developed by the Wallingford Centre for Ecology and Hydrology.

### 2.5. Methods for Development of the Management Platform

The integrated urban flooding early warning and rainfall runoff management platform developed in this study is based on an off-line drainage network model, data integration, and a real-time model system on which the on-line software system is executed. The drainage facility GIS is the data source of the network used for model establishment, the off-line model is the major engine for model calculation and simulation, and the on-line model platform provides an environment that allows automatic operation. Within the context of user-oriented design, a web client real-time publishing system was developed with a user-friendly and easy-to-operate interface. The main methods and procedures adopted for developing the management platform are outlined below.

#### 2.5.1. Development of the off-line drainage network model

The technical roadmap for platform implementation is shown in Figure 3.

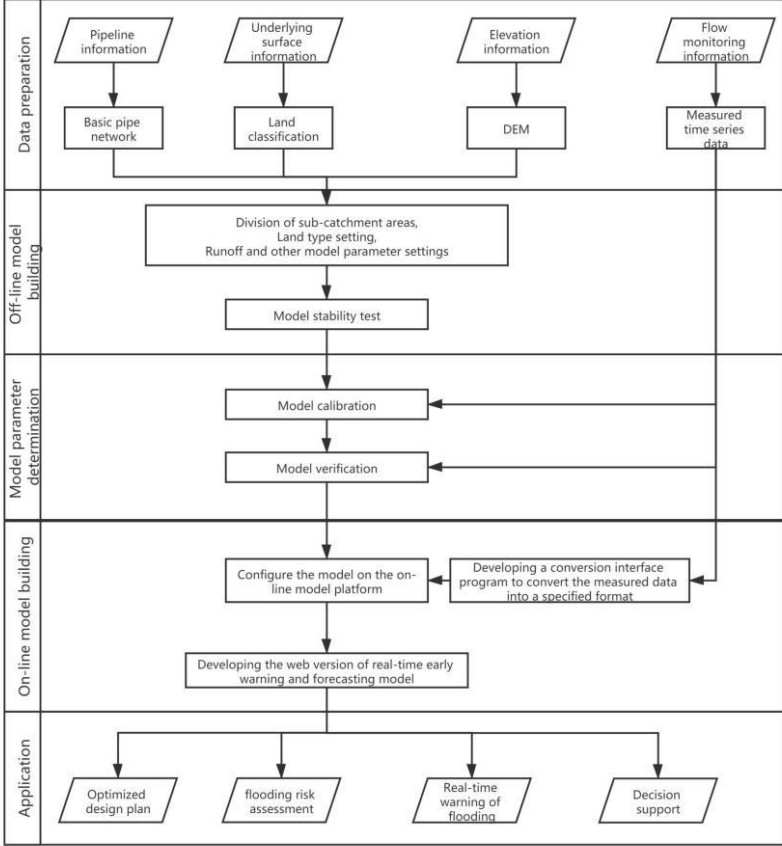

**Figure 3.** Technical roadmap for platform implementation.

### 2.5.2. Development of the Off-Line Drainage Network Model

(1) Collect information related to inspection shafts, pipelines, pumping stations, catchment area, and supporting drainage facilities within the research area, and check and verify the accuracy of the information;

(2) Collect information related to urban spatial attributes within the research area, e.g., terrain elevation data, land use composition data, and RS imagery;

(3) Collect dry/rainy day field measurements for model calibration and verification;

(4) Analyze the underlying surface and categorize the land use within the research area into four types: buildings, roads, greenbelt, and other;

(5) Develop 1- and 2-dimensional drainage system models and calibrate and verify the models based on field measurements. Grid division for the 2-dimensional model was achieved based on a triangulated irregular network with major consideration given to the terrain and landscape features of the research area, e.g., watercourses, water bodies, roads, and buildings, where buildings were treated as clear areas and thus not involved in the 2-dimensional flooding process. With an average grid area of 25–100 m$^2$, the research area was divided into 620,000 grids. By taking into account the impact of land features such as the underlying surface, presence of buildings, and ground elevation on the movement of floodwater, the 2-dimensional model can simulate surface flooding and recession with improved accuracy. History rainfall data as shown in Table 1 were used to validate and verify the model. Runoff parameters, underground seepage, coefficient of roughness, head loss, etc., of the model were adjusted to allow the model to reflect the actual hydraulic characteristics. According to model validation and verification results, the overall flow deviation of the system is within 10%, the water level simulation curve for the intake structure fits relatively well with the measurements curve, and the 2-dimentional flooding spatial location and depth are in line with the actual flooding conditions. Therefore, the model developed is able to meet the requirements;

(6) Assess the capacity of the drainage system within the research area, and work out measures and schemes for drainage capacity optimization. Characterized by low and flat terrain and low drainage capacity, the research area is subject to flooding and the underground space is subject to water ingression when the rainstorm exceeds the design standard. Given its position as the CBD center of Shanghai government and Southwest region, flooding will pose an even more severe impact to social economic development.

### 2.5.3. Development of On-Line Real-Time early Warning and Forecasting System

(1) Development of real-time RS data conversion program

The system automatically and regularly collects rainfall field measurements and drainage facility operating data, and it stores such information in the database. To meet the requirements of the FloodWorks software in terms of data format, a data interface conversion program was developed to realize the format conversion of the field measurements to facilitate automatic operation.

(2) Development of real-time early warning and forecasting system

The drainage facility GIS database integrates real-time RS data of the SCADA system, e.g., rainfall, water level, and water pump on/off status at relevant monitoring points within the research area. Configured with the FloodWorks server, and based on read-in measurements and the drainage network model, the system responds to an imminent rainstorm according to current operating procedures. Based on the preset scheduling rules, the system enables automatic model simulation and future strategies for system operation and pumping station scheduling, including how many and at what time the pumps should be activated.

### 2.5.4. Development of Web Client Real-Time Publishing System for the Real-Time Model System

(1) Procedures for migrating calculation results file of the model

To accelerate the speed at which the web data are read and to realize rapid inquiries regarding historical simulation results, the calculation results of the model are stored in the

database. Through programming, the model data read by the web system are stored in the Oracle database of the system via software tools.

(2) Development of real-time web client publishing system

With a user-friendly design, and based on SCADA real-time data measurements and model calculation results, the web client adopts ArcGIS web publishing technology to dynamically display the status of the operating conditions of the pumping stations and information regarding the flood monitoring points within the forecasted area. Thus, management personnel and the public are provided with data relating to potential areas of flooding and forecasted flooding and recession times.

## 3. Results and Discussion

### 3.1. Major Design Features of the Management Platform

(1) Operation browsing feature

The FloodWorks software interface and the web publishing system of the real-time system both allow browsing of SCADA-based data charts and trilinear charts (e.g., rainfall, water level, pump on/off status) of the pumping stations (Figure 4). The information pertaining to rainfall, water level, and pump on/off status also serves as the basis for dynamic calculation and updating the model.

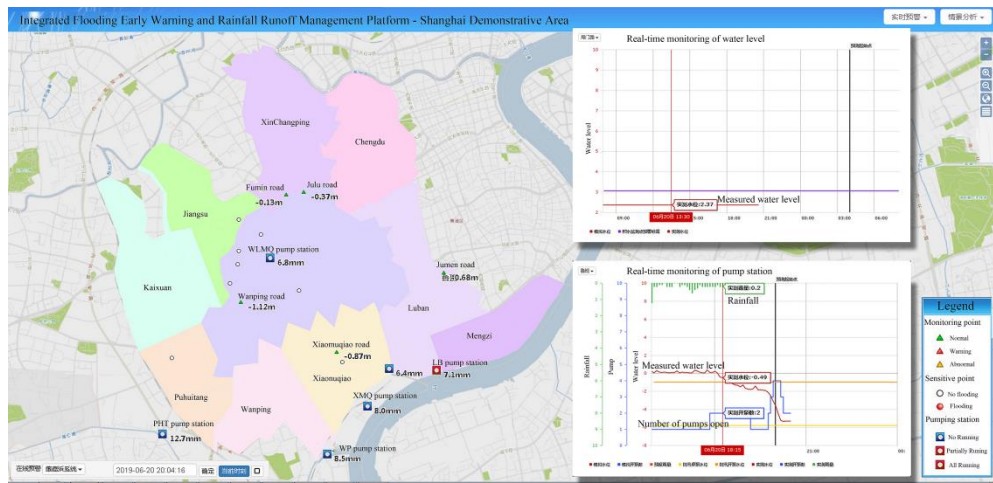

**Figure 4.** Web client operation browsing interface for the management platform.

(2) Forecasting and early warning feature

The drainage network hydrodynamic model embedded in the real-time platform serves as the computing engine of the simulation and forecasting system. Based on the relative static characteristics of the network system, the forecasting system responds to an imminent rainstorm according to historical operating conditions, and it provides system operation adjustment and strategies for pumping station scheduling according to preset scheduling rules. Existing pumping station scheduling functions and resources are the foundation for the realization of a scheduling strategy, and existing drainage system monitoring conditions are prerequisite for model development and real-time scheduling. The real-time modeling system developed in this study provides two forecasting modes: manual forecasting and automatic forecasting.

The former allows forecasting of the total amount and duration of rainfall expected in the following few hours according to a preset rainfall pattern. The latter allows automatic forecasting of rainfall data, which depends on the availability of automatically updated radar data. Radar technology enables measurement of rainfall within a certain radius (e.g., 200 km) around the pumping stations, with resolution of up to 1 km based on terrain characteristics. Currently, the two radar stations in Shanghai, which are located in Pudong District and Qingpu District, allow real-time publishing of radar precipitation images. In this study, through cooperation with the Shanghai Meteorological Observatory, and based

on analysis of the gridded radar forecasting results, analysis and integration of real-time radar data was realized.

The on-line real-time model can predict the following information within the research area through simulation and calculation:

Location, depth, and duration of flooding within the serviced area;

Change of water level in the intake structure and discharge volume of the pumping stations;

Dynamic process of surface flooding within the serviced area.

The calculation results can be browsed on-line through either the FloodWorks software interface or the interface of the web publishing system of the real-time model system.

(3) Scheduling support feature

The integrated urban flooding early warning and rainfall runoff management platform mainly provides the following three elements of real-time scheduling support for drainage facilities.

First, it provides support for flooding mitigation plans and countermeasures based on flooding forecasts. Based on the location, depth of flooding, and recession time forecasted by the real-time model, the platform develops emergency evacuation plans and temporary pump truck deployment plans in advance.

Second, it provides support for pumping station operation scheduling according to the forecasted results. For example, a low-water-level operation mode should be adopted prior to the imminent arrival of a forecasted rainstorm to meet the flooding control requirements. In the case of forecasted heavy rain and rainstorms, or manual input of heavy rain and rainstorms, the real-time model will simulate the number of on/off pumps and the timings for pump operation to control the water level of the drainage system in accordance with the measured/forecasted rainstorm conditions, ensuring flooding control safety over the subsequent few hours. With the update of real-time data, the platform will calculate and update the forecasted results at 15-min intervals. In turn, the corresponding pump on/off status will be updated and this cycle continues. Based on real-time data, this pumping station scheduling mode takes account of the overall flooding control safety, ensuring precise and scientific operation and scheduling of the pumping stations.

Third, it determines the optimal scheme according to comparative results. The platform provides a tool that allows comparison of the impact of different rainfall conditions and different combinations of pumping station status on the operation of the drainage system. It displays the comparative information that includes the number of pumps in operation, number of flooding points, discharge volume to the river, and peak flow under different scenarios for analysis. It allows scheduling personnel to determine the optimal scheduling scheme by switching the preset priority settings between reducing combined sewer overflows discharge and ensuring flooding control safety. The related diagram is showed in Figure 5.

(4) Web publishing

The web publishing of the forecasted results is intended to disseminate the simulated rainstorm flooding, drainage network, and pumping station operating results to the target group. Based on web–GIS technology, a Brower/Server information system was established that allows for the dynamic display of 2- and 3-dimensional flooding and recession charts on the webpage. Web publishing allows real-time inquiry and display of statistics of the basic data and operating data of the infrastructure.

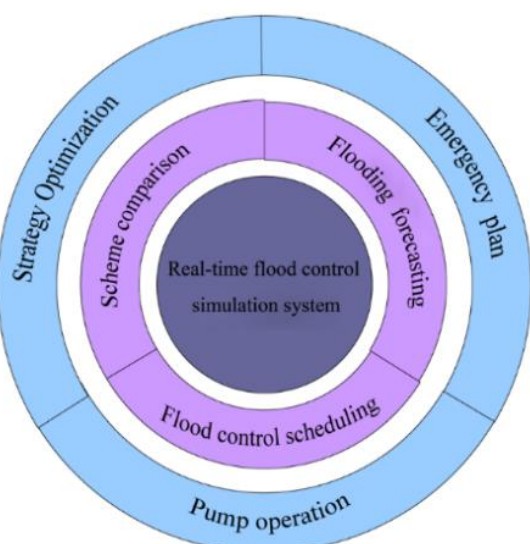

**Figure 5.** Real-time scheduling support feature of the management platform.

### 3.2. Operating Mechanism Design of the Management Platform

The management platform offers three operating modes that can be switched according to different operating conditions.

(1) Off-line mode

The off-line mode enables technical personnel to conduct model analysis and real-time system construction. Integrating a wide range of information, the off-line mode is intended mainly for detailed analysis and updating of the models and scheduling schemes.

(2) On-line dry day mode

On dry days, this mode runs automatically in the background to provide hot-start data for the on-line models. At this point, the pumping stations are scheduled according to the routine operational mode. The platform mainly provides trilinear charts associated with the actual operating conditions for browsing (i.e., rainfall, pump on/off status, water level of the intake structure).

(3) Rainy day early warning mode

On rainy days, the management platform provides forecasting progress, and it generates scheduling schemes and operation reports that can be used to develop pumping station scheduling and emergency plans.

(4) Mode switching mechanism

The off-line mode is used for model analysis as well as platform construction and updates. After the real-time model is published, the on-line routine mode will be activated. Relevant personnel can view the actual operating conditions of the pumping stations within the research area or initiate forecasting through a local area network or the Internet. Once local rainfall occurs, the RS rainfall data will be transmitted automatically to the real-time platform and the system will activate the on-line early warning mode. The on-line early warning mode can also be activated by scheduling personal through manual input of rainfall forecasts. The real-time platform is capable of providing a recommended process route for flooding control and pumping station operation in the following few hours based on the rainfall process according to preset low-water-level control rules. Within hours of the cessation of the rainfall, the real-time platform will automatically switch back to its routine operational mode.

### 3.3. Application Performance of the Management Platform

(1) Model calibration and verification

Choose four rains of different duration and intensity that occurred in 2010–2016, and use actual rainfall process and basic dry flow as input to calibrate and verify the model. The calibration results of the water level and outflow in the front pool of the pumping

station are good. The average Nash–Sutcliffe coefficients are 0.75 and 0.71, respectively. The relevant curve of a typical pumping station is shown in Figure 6.

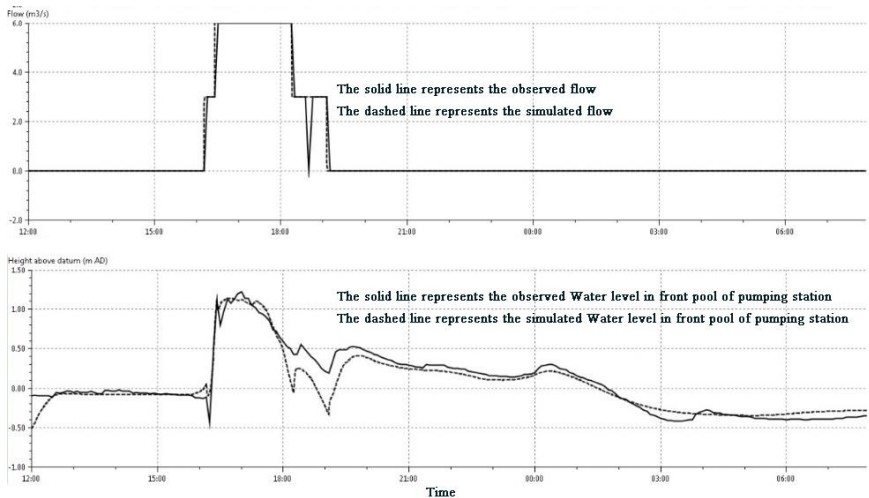

**Figure 6.** Relevant curve of a typical pumping station.

(2) Drainage capacity and risk assessment results

The drainage system selected in this study was designed based on one-in-one-year flood. The off-line hydrodynamic model, which allows for evaluation of the rainstorm flooding risk, is capable of checking whether the drainage system of the area meets the design capability, as well as providing a sound basis for the development of emergency early warning plans.

The designed rainfall includes two elements: rainfall intensity and duration. Shanghai City has conducted validation on applicable rainstorm intensity formula and design rainfall pattern [12], which have been published in Shanghai local standards [13]. In this study, the short-term rainstorm intensity was assessed using the Chicago rainfall pattern stipulated by Shanghai local standards. The rainfall duration was selected as two hours and the location coefficient of the rainfall peak was selected as 0.405. The return periods of the rainstorm were selected as one-in-one-year, one-in-two-year, one-in-three-year, and one-in-five-year. The Shanghai rainfall process diagram with different return periods based on Chicago rainfall patterns is shown in Figure 7.

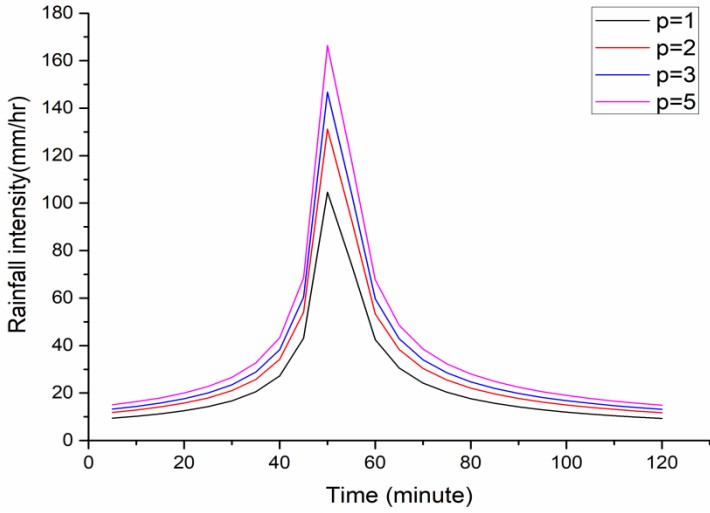

**Figure 7.** Shanghai rainfall process with different return periods based on Chicago rainfall patterns.

The off-line drainage network hydrodynamic model was used to evaluate the flooding risk associated with rainstorms within the research area. The system drainage capacity was quantified in terms of four indicators: flooded road ratio, flooded area ratio, total amount

of flooding, and average flooding time (Figures 8 and 9). The results indicate that the drainage system within the research area could meet the one-in-one-year design standard. However, the flooding risk is relatively high for extremely heavy rainfall. The indicators used to quantify drainage system capacity were defined as follows:

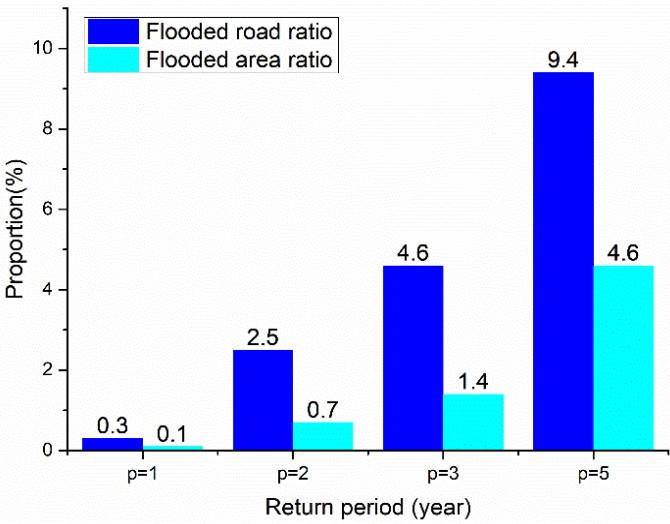

**Figure 8.** Bar chart of flooded road ratio and flooded area ratio.

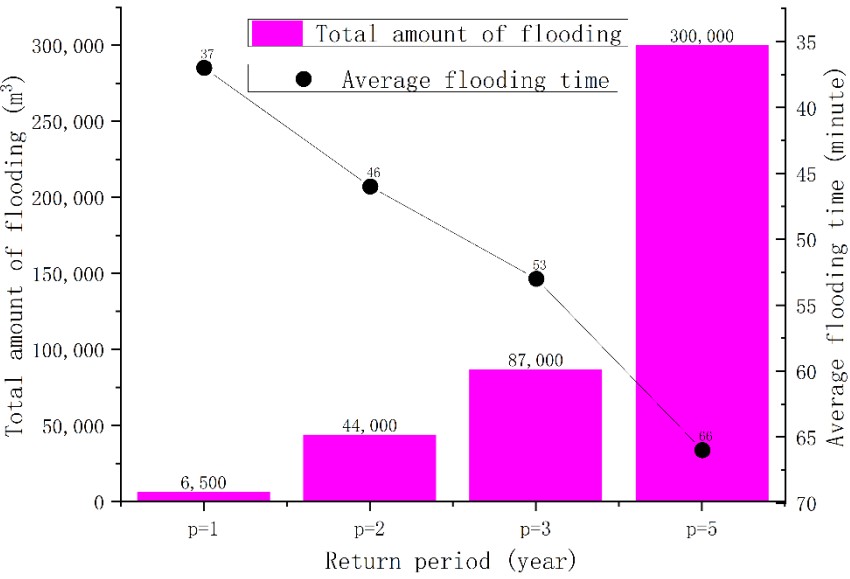

**Figure 9.** Bar chart of total amount of flooding and average flooding time.

i. Flooded road ratio: the ratio between roads flooded to a depth of >0.15 m and all roads;

ii. Flooded area ratio: the ratio between the area flooded to a depth of >0.01 m and the total area within the service area;

iii. Total amount of flooding: the total amount of flooding to a depth of >0.01 m within the flooded area;

iv. Average flooding and recession time: the average duration of the flood with depth >0.15 m.

(1) Operation status of online real-time early warning and forecasting system

Since its completion and application in 2016, the real-time flooding management model system has delivered outstanding performance in the subsequent flooding seasons and it has realized continuous real-time operation. The performance of the on-line real-time early warning and forecasting system is illustrated based on the rainfall event that occurred

in Shanghai on 4 August 2016. The total rainfall of this event was 74.3 mm, the maximum hourly rainfall intensity was 38.9 mm/h (which exceeded the one-in-one-year flood design standards, and the overall duration was approximately three hours. The rainfall process curve is shown in Figure 10.

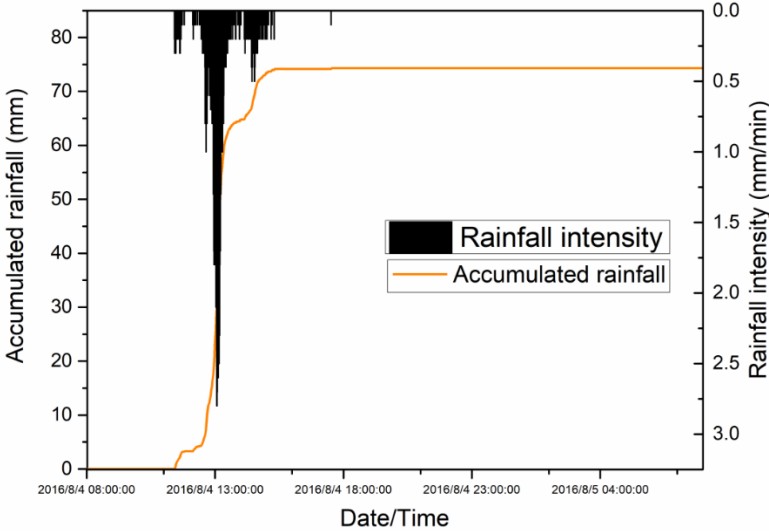

**Figure 10.** Rainfall process of the event on 4 August 2016.

For this rainfall event, the management platform generated 23 early warnings and forecasts and it provided 23 recommended scheduling schemes. Based on the real-time rainfall data, the platform automatically simulated and provided pumping station scheduling schemes for the subsequent four hours, which provided a reference for pumping station scheduling optimization. Here, a web-page screenshot of the simulation results captured at 13:30 local time is used for illustration (Figure 11). Upon clicking the area after the forecast starting point, the web client will automatically display the number of pumps that should be activated and the pump start times as recommended by the simulator, providing the operating personnel with a reference for real-time operational mode adjustment.

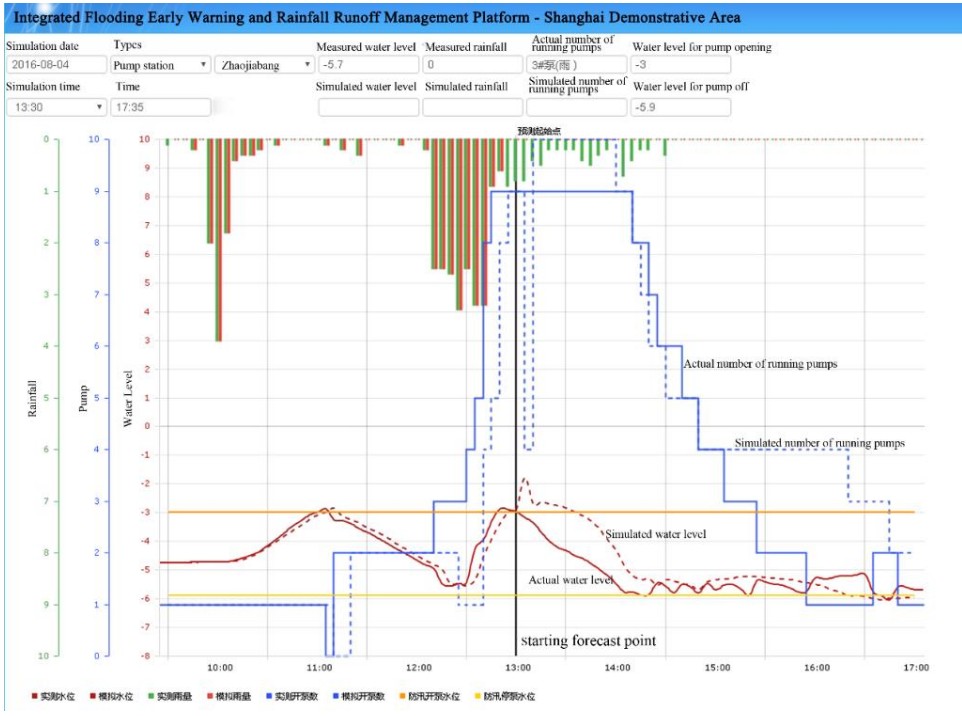

**Figure 11.** Analysis interface for pumping station real-time scheduling optimization.

(2) Analysis of Factors Affecting Simulation Accuracy

According to the experience accumulated during the implementation of this platform, there are two main factors that affect the accuracy of the simulation. The first is the accuracy of the weather forecast. The forecast accuracy of the rainfall process in the next few hours by the meteorological department will have a great impact on the simulation results of flooding. The second is the accuracy of the urban spatial data such as the pipeline network. The model needs to be maintained regularly to reflect the actual pipeline network situation.

## 4. Conclusions

Within the context of the "Smart Water" initiative in Shanghai, through integration of a wide range of data from various sources, and using a computing engine based on one- and two-dimensional drainage network models and real-time early warning and forecasting models, this study established an integrated urban flooding early warning and rainfall runoff management platform for Xujiahui District in downtown Shanghai. The advantage of the platform is that based on the weather forecast, the platform can deal with hidden flood hazards in advance and formulate response plans in advance. The disadvantage is that it has higher requirements for urban infrastructure data, high requirements for the engineers' level of model construction and maintenance, and higher capital investment. The platform realized the construction and application of the drainage system real-time warning and forecast platform for the first time in China. Due to the repeated occurrence of urban flooding incidents in recent years, the city management department is promoting the development of a platform that serves a larger area, and the research can provide technical support for it.

As one of the components of the integrated urban flooding early warning and rainfall runoff management platform, a real-time drainage model system is rarely adopted in practice in similar systems used around the world. As a project that demonstrates the Water Pollution and Control component of China's "Twelfth Five-Year Plan," this project has transformed from a simple experience-based management approach to a sophisticated management platform based on real-time data and scientific calculation. Presenting an effective approach by Shanghai to the implementation of the "Smart Water" initiative, this study not only provides technical support for flooding risk prevention and control in Shanghai, but it also serves as a demonstration of an urban flooding real-time early warning system that could be adopted throughout China.

**Author Contributions:** Conceptualization, Z.S. and T.L.; Data curation, Q.S. and Q.T.; Methodology, Z.S. and T.L.; Project administration, Z.S. and Q.T.; Software, Q.T. and Q.S.; Validation, Q.T.; Writing—review & editing, Q.S., Z.S. and T.L. All authors have read and agreed to the published version of the manuscript.

**Funding:** i. National Major Science and Technology Project of Water Pollution Control and Management in the 12th Five Year Plan, 2013ZX07304-003-06 ii. Scientific Research Project of Shanghai Scientific and Technological Committee, 14DZ1208200.

**Institutional Review Board Statement:** Not applicable.

**Informed Consent Statement:** Not applicable.

**Conflicts of Interest:** The authors declare no conflict of interest.

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
