# Peer review of "Development of Integrated Flooding Early Warning and Rainfall Runoff Management Platform for Downtown Area of Shanghai"

_sustainability, doi:10.3390/su132011250_

Round 1

Reviewer 1 Report

Journal: Sustainability (ISSN 2071-1050) Manuscript ID: sustainability-1398314 Type: Article Title: Development of Integrated Flooding Early Warning and Rainfall Runoff Management Platform for Downtown Area of Shanghai   Comments to Authors,   The manuscript was developed with appropriate Figures and Tables. The Abstract, Introduction, Conclusions sections were explained clearly.     I recommend authors to do following modifications to consider the manuscript for the publication.   Mention the aim of the study clearly at the end of the introduction section.   Mention obtained results under a new sub topic as "Results" or combine results section with Discussion section as "Results and Discussion". In this manuscript the "Results" section is totally absent.    

Author Response

Response to Reviewer 1 Comments

Point 1: Mention the aim of the study clearly at the end of the introduction section.

Response 1:

The purpose of the research is added to the end of the introduction:

This platform will not only improve the capacity of urban drainage networks to guard against flooding, but also provide a support for drainage network scheduling, flooding early warning, and flood hazard mitigation. On the one hand, the platform can enhance features that include flooding early warning, forecasting, and risk assessment based on numerical models, providing technical support for the safety of drainage and flooding in the study area. On the other hand, the platform has formed a set of key technologies for the construction of a real-time warning model for drainage systems, and provided a blueprint for the construction of similar flood control platforms in other medium-sized and above cities and towns.

Point 2: Mention obtained results under a new sub topic as "Results" or combine results section with Discussion section as "Results and Discussion". In this manuscript the "Results" section is totally absent.

Response 2:

Revise the discussion to result and discussion, which is more in line with relevant content

Reviewer 2 Report

The study is well-prepared and deals with the important issue of development of integrated flooding early warning and rainfall runoff management platform for downtown area of Shanghai. The study used data obtained from an urban drainage network and spatial geological information to conduct precise analysis on an area of approximately 31 km2 with various land use in downtown Shanghai and to establish a 2-dimensional model.

Remarks: Are there concrete steps that can be recommended and how generalizable are the findings? The region in which the inquiry was conducted, What's distinctive about it? Please indicate what new research brings. The advantages and disadvantages of the proposed method should be indicated. Its limitations and generalized application should be presented. Is this approach was consulted with water and sewer managers? References should be numbered in the text and prepared according to the journal’s guidelines.

Reviewer 3 Report

Thank you for allowing me to review this article.

The manuscript entitled "Development of Integrated Flooding Early Warning and Rainfall Runoff Management Platform for Downtown Area of Shanghai" is an original contribution, and the topic is interesting for readers of the Sustainability journal.

The presentation is fine but I noticed criticism in the text that should be addressed accordingly. Authors can use the comments below to improve their article.

1) I suggest adding the coordinates of the study area (in text or on figure 1).

2) Line 124 is blank - delete.

3) Table 2 should be before Figure 2..

4) The section "Introduction" is fine. However, I suggest adding more references.

Author Response

Response to Reviewer 3 Comments

Point 1: I suggest adding the coordinates of the study area (in text or on figure 1).

Response 1: The area is located at 121.45 north latitude and 31.2 east longitude. The sentence is added to the text in SCOPE OF STUDY section.

Point 2: Line 124 is blank - delete.

Response 2: Modified as required.

Point 3: Table 2 should be before Figure 2.

Response 3: Modified as required.

Point 4: The section "Introduction" is fine. However, I suggest adding more references.

Response 4: Modified as required.

Reviewer 4 Report

The manuscript provide an report on the development of integrated flooding early warning and rainfall runoff management platform for downtown area of Shanghai. The topic is important as many cities are facing flood, and it is a big problem in many cities in the world. The paper brings good concept and results to assess the drainage system and early warning of flooding risk. However, the presentation of the manuscript is not scientific sounds, rather than a general report, there are some points need to be correct before it can be considered for publication.

  • Introduction should be rewritten. It should be expanded to include a more detailed discussion of current problems, and the limitation of various systems for urban drainage network management and real-time monitoring and early warning and forecasting models.
  • Novelty of the study, the authors need to point out their objectives clearly and novelty of the study, which need to be stressed.
  • 4. Selection of modeling software platform, this part is not convience, the authors need to compare the different softwares in a clear way, and why it’s suitable for this study area.
  • Discussion must change to Results and Discussion.
  • Critical comments should be made on the results of the cited works.
  • The discussion statements are speculations. A more detailed discussion of factors affecting the observed performance should be added. Make every attempt to improve the discussion by critically analyzing your findings.
  • Very important for the results and discussion part is model verification and the accuracy of the model, this part is missing, which need to be add in the revised version.
  • Conclusions should be amended to incorporate a broader discussion of the significance and potential application of this specific study. More concise and direct conclusion is appreciated.
  • The current English language is too colloquial and hard to understand, a more scientific language is needed. The use of English language needs to improve by a native speaker.

Author Response

Response to Reviewer 4 Comments

Point 1:. Introduction should be rewritten. It should be expanded to include a more detailed discussion of current problems, and the limitation of various systems for urban drainage network management and real-time monitoring and early warning and forecasting models.

Response 1: In China, due to the rapid urban development and climate change, extreme meteorological events are occurring more frequently.For example, on July 20th of 2021, more than 200 mm of rain fell on the city of Zhengzhou in a single hour, in China's Henan province. The precipitation accumulated up to 552 mm in 24 hours (breaking the historical extreme records), caused huge loss of personnel and property.However, the current measures to deal with flood disasters are relatively single, and there is a lack of technical means to forecast and warn the areas where serious flooding disasters occur. There is a large gap between the accuracy of disaster early warning and the requirements of safety assurance. It is necessary to improve the ability of flood prevention and disaster mitigation early warning and forecasting and the ability to deal with major flood disasters. In particular, it is necessary to develop a multi-disciplinary, cross-domain, and multi-department integrated management and control platform.

Therefore, the development of a system for urban drainage network management based on real-time monitoring and on an early warning and forecasting model is the main trend in current research for urban flooding prevention and management. The implementation of this project fills up the gaps in the application of this field in Chinese cities and towns.

This platform will not only improve the capacity of urban drainage networks to guard against flooding, but also provide a support for drainage network scheduling, flooding early warning, and flood hazard mitigation. On the other hand, the platform has formed a set of key technologies for the construction of a real-time warning model for drainage systems, and provided a blueprint for the construction of similar flood control platforms in other medium-sized and above cities and towns.

This sentence is added to the end of INTRODUCTION section.

Limitations were discussed in the conclusion section.

Point 2: Novelty of the study, the authors need to point out their objectives clearly and novelty of the study, which need to be stressed.

Response 2: The research has formed the key technology for constructing the flooding prevention early warning forecast model for the drainage system of Chinese cities and towns. Mentioned in the response 1.

Point 3:. 4. Selection of modeling software platform, this part is not convince, the authors need to compare the different softwares in a clear way, and why it’s suitable for this study area.

Response 3: The off-line hydrodynamic modeling software platforms used most often in previous studies include SWMM, InfoWorks CS,  InfoWorks ICM and DHI urban, and the most commonly used on-line real-time early warning modeling software platforms include FloodWorks and MIKE Flood[Gong et al 2012]. SWMM cannot simulate two-dimensional flooding, DHI series software is more inclined to watershed simulation.This project needs to carry out two-dimensional real-time simulation of flooding at the level of urban drainage system, so the InfoWorks series software was selected as the modeling software platform.

This sentence is added in the  Selection of modeling software platform.

Point 4:. Discussion must change to Results and Discussion.

Response 4: Modified as required.

Point 5:.Critical comments should be made on the results of the cited works.

The discussion statements are speculations. A more detailed discussion of factors affecting the observed performance should be added.

Response 5:

(4)Analysis of Factors Affecting Simulation Accuracy

According to the experience accumulated during the implementation of this platform, there are two main factors that affect the accuracy of the simulation. The first is the accuracy of the weather forecast. The forecast accuracy of the rainfall process in the next few hours by the meteorological department will have a great impact on the simulation results of flooding. The second is the accuracy of the urban spatial data such as the pipeline network. The model needs to be maintained regularly to reflect the actual pipeline network situation.

This sentence is added in the 3. Application performance of the management platform .

Point 6:Very important for the results and discussion part is model verification and the accuracy of the model, this part is missing, which need to be add in the revised version.

Response 6: Modified as required.

Point 7:Conclusions should be amended to incorporate a broader discussion of the significance and potential application of this specific study. More concise and direct conclusion is appreciated.

Response 7: The advantage of the platform is that based on the weather forecast, the platform can deal with hidden flood hazards in advance and formulate response plans in advance. The disadvantage is that it has higher requirements for urban infrastructure data, high requirements for the engineers’ level of model construction and maintenance, and higher capital investment. The platform realized the construction and application of the drainage system real-time warning and forecast platform for the first time in China. Due to the repeated occurrence of urban flooding incidents in recent years, the city management department is promoting the development of a platform that serves a larger area, and the research can provide technical support for it.

This sentence is added to the CONCLUSIONS.

Point 8:The current English language is too colloquial and hard to understand, a more scientificlanguage is needed. The use of English language needs to improve by a native speaker.

Response 8: The author rechecked the grammar and revised the main parts such as the conclusion.

Round 2

Reviewer 1 Report

Journal: Sustainability (ISSN 2071-1050) Manuscript ID: sustainability-1398314 Type: Article Title: Development of Integrated Flooding Early Warning and Rainfall Runoff Management Platform for Downtown Area of Shanghai   Comments for authors:   Abstract, Materials and Methods, Results and Discussion and Conclusions sections were well developed with appropriate figures and tables.

Reviewer 4 Report

The revised manuscript stressed all my comments, I have no further comments.